# DPF-Nutrition: Food Nutrition Estimation via Depth Prediction and Fusion

**DOI:** 10.3390/foods12234293

**Published:** 2023-11-28

**Authors:** Yuzhe Han, Qimin Cheng, Wenjin Wu, Ziyang Huang

**Affiliations:** 1School of Electronic Information and Communication, Huazhong University of Science and Technology, Wuhan 430074, China; 2Institute of Agricultural Products Processing and Nuclear Agricultural Technology, Hubei Academy of Agricultural Science, Wuhan 430064, China

**Keywords:** nutrition estimation, deep learning, depth prediction, RGB-D fusion

## Abstract

A reasonable and balanced diet is essential for maintaining good health. With advancements in deep learning, an automated nutrition estimation method based on food images offers a promising solution for monitoring daily nutritional intake and promoting dietary health. While monocular image-based nutrition estimation is convenient, efficient and economical, the challenge of limited accuracy remains a significant concern. To tackle this issue, we proposed DPF-Nutrition, an end-to-end nutrition estimation method using monocular images. In DPF-Nutrition, we introduced a depth prediction module to generate depth maps, thereby improving the accuracy of food portion estimation. Additionally, we designed an RGB-D fusion module that combined monocular images with the predicted depth information, resulting in better performance for nutrition estimation. To the best of our knowledge, this was the pioneering effort that integrated depth prediction and RGB-D fusion techniques in food nutrition estimation. Comprehensive experiments performed on Nutrition5k evaluated the effectiveness and efficiency of DPF-Nutrition.

## 1. Introduction

Dietary health has become the predominant focus in modern life. Excessive or imbalanced intake may lead to different kinds of diet-related diseases, especially obesity, which will dramatically increase the risk of hypertension, cardiovascular disease and diabetes [1]. Misestimating of nutrition content is a significant factor in excessive and imbalanced intake. The International Food Information Council (IFIC) Foundation reported that most people tend to overestimate their vegetable intake while underestimating their fat intake [2]. Therefore, there is an urgent demand for effective nutrition estimation methods to help individuals monitor their daily dietary intake and guide them to a healthier diet. Previous dietary assessment methods heavily relied on human involvement. Specifically, the commonly used 24 h Dietary Recalls [3] require participants to report their food types and portion sizes over a 24 h period, thereby understanding their eating behavior. Many popular applications, i.e., MyFitnessPal, MyDietCoach, Yazio, FatSecret, MyFoodDiary and Foodnotes, are all developed based on this method. Despite the advantage of being easy to implement, it is burdensome and unreliable due to its high dependence on the subjective judgements of the participants.

Fortunately, recent development in Artificial Intelligence (AI), especially in deep learning techniques, have made the automated and reliable dietary assessment a reality [4,5]. Vision-based nutrition estimation methods allow users to monitor their food intake by capturing images using their mobile devices, which heavily reduces user burden. According to the type of input data, existing methods can be broadly divided into three categories [6]: monocular image-based methods, multi-view image-based methods and RGB-D methods. Earlier works [7,8] have predominantly relied on multi-view images to reconstruct the 3D structure of food objects and estimate their volume. Subsequently, they calculated the sum of nutrition by combining the volume with the nutritional information of the food. Nevertheless, multi-view image methods are troublesome and inefficient since they require users to capture images from specific angles. In contrast, monocular image-based methods only relied on a single food image and demonstrated good performance. Specifically, Shao et al. [9] proposed a method for estimating calories based on an energy density map, which maps RGB images to the energy density of food on a pixel-to-pixel basis. They compared the proposed method with the manual 24 h Dietary Recalls and the results showed an obvious advantage. Similarly, Thames et al. [10] demonstrated that the performance of the vison-based method outperformed the professional human nutritionists where they used a multi-task convolutional neural network to estimate multiple nutrients. Shao et al. [11] employed a combination of non-destructive detection technology and deep learning to analyze the nutritional content of food. They improved the detection of small target foods to improve the nutrient estimation accuracy. However, estimating food nutrition from a monocular image is an ill-posed problem [12]. This is because the process of mapping food objects to monocular images often leads to a loss of crucial 3D information, which is vital for food portion estimation. In order to resolve the problem, depth information is utilized to complement the 3D information lost in monocular images. Lu et al. [13] utilized RGB-D pairs captured from real eating scenarios as input, and integrated techniques of food segmentation, recognition and 3D surface reconstruction to estimate nutrient intake for hospitalized patients. Thames et al. [10] incorporated the monocular images with depth maps as four-channel data, which were subsequently sampled into a three-channel tensor as the input of the model. However, these methods only treated food images and depth maps as image pairs, neglecting the inherent differences between RGB and depth images, which limits the performance of nutrition estimation. Recently, cross-modal fusion has been explored to enhance the performance of nutrition estimation. Vinod et al. [14] employed normalization techniques to address disparities in the feature space between the energy density map [9] and the depth map. Shao et al. [15] employed the balanced feature pyramid [16] and the convolutional block attention module [17] to enhance the fused features, but the fusion of RGB and depth features still employed a straightforward concatenation approach. These methods have facilitated the research on RGB-D nutrition estimation, primarily through their focus on modal disparity. However, the enhancements failed to explore the interactive effect of cross-modal features, thereby restricting the extraction of complementary information that could further enhance the accuracy of nutrition estimation. In addition, the acquisition of a depth map heavily relies on professional depth sensors, which increases the cost and restricts the application scenarios of RGB-D nutrition estimation methods.

In this paper, we proposed a novel food nutrition estimation method based on Depth Prediction and Fusion, referred to as DPF-Nutrition. We employed a depth prediction model that generated predicted depth maps instead of actual depth maps captured by depth sensors. These predicted depth maps recovered the missing 3D information in monocular images, thereby enhancing the accuracy of food portion estimation without additional cost. Unlike existing RGB-D fusion approaches in nutrition estimation [14,15] that enhanced features on a single modality, we proposed a Cross-modal Attention Block (CAB) that focused on the interaction effect of the cross-modal features. The CAB utilized cross-modal attention features to enhance single-modal features, leading to increased complementarity between modalities and the generation of more discriminative fused features. This improvement allows the model to accurately focus on the correct nutrient regions. In addition, we proposed a multi-scale fusion network that enhanced the semantic features through combining the fused features at different resolutions, thereby enabling the model to capture the co-occurring food features in the feature maps. The proposed CAB and multi-scale fusion network consist of the RGB-D fusion module in DFP-Nutrition. This module effectively exploits the features of both RGB and depth images to enhance the performance of food nutrition estimation. During the inference stage, DPF-Nutrition relies solely on monocular food images as input, making it essentially a monocular image-based method. We evaluated the effectiveness of DPF-Nutrition on the public Nutrition5k dataset and obtained encouraging results. Specifically, the percentage mean absolute errors (PMAEs) for calories, mass, protein, fat and carbohydrate estimation reached 14.7%, 10.6%, 20.2%, 22.6% and 20.7%, respectively. Compared with the previous monocular image-based method proposed by Thames et al. [10], the mean PMAE of our DPF-Nutrition reached 17.8%, improved by 11.3%. Furthermore, when compared with the state-of-the-art RGB-D method proposed by Shao et al. [15], our DPF-Nutrition demonstrated competitive performance with a 0.7% improvement.

The contribution of this paper can be summarized as three aspects:We proposed a novel monocular image-based nutrition estimation method based on Depth Prediction and Fusion, referred as DPF-Nutrition. It was the first attempt to incorporate depth prediction and RGB-D fusion techniques in vision-based nutrition estimation.We designed an RGB-D fusion module that incorporated the proposed cross-modal attention block (CAB) and multi-scale fusion network to fully exploit the informative image features for nutrition estimation.Our proposed DPF-Nutrition demonstrated effectiveness in accurately estimating multiple nutrients, which has been evaluated on the public dataset Nutrition5k.

## 2. Materials and Methods

### 2.1. Dataset

We summarized diversity, size, annotations and inclusion of depth information in nutrition datasets. The result is shown in Table 1. Overall, the Nutrition5k dataset [10] contained the highest number of unique dishes and images. Furthermore, it contained extensive nutritional annotations, whereas other datasets only included single calorie or portion annotations. In this paper, we evaluated our DPF-Nutrition method on the Nutrition5k dataset. This dataset comprises 20k short videos and 3.5k RGB-D images captured by an Intel RealSense camera, involving approximately 5k distinct food dishes. Each dish within the dataset includes detailed information such as ingredient names, quantities and associated macronutrient data calculated using the reliable USDA Food and Nutrient Database [18]. Figure 1 demonstrates examples of images in the Nutrition5k dataset. In the depth map, objects that are closer to the camera are represented in blue, while objects that are farther away are shown in red. The color bar serves as a visual indicator, reflecting the varying distances from each point to the camera, with the unit of measurement in centimeters. As an image-based nutrition estimation method, our DPF-Nutrition was evaluated on the 3.5k food images from the Nutrition5k dataset. The Nutrition5k dataset provides a predefined split, dividing the data into training and testing subsets with a ratio of 5:1. There exists a small number of erroneous images, as shown Figure 2, which have the potential to mislead the training of the model, resulting in suboptimal performance. However, in order to maintain consistency in the comparison of methods in this research, we have chosen to preserve the integrity of the dataset and refrain from performing any additional data cleaning.

### 2.2. Methods

The overall architecture of our DPF-Nutrition is illustrated in Figure 3. The model comprises two main modules:**Module1: Depth prediction module** aims to reconstruct the 3D depth information based on 2D monocular images. The depth prediction module employs a vision transformer as the encoder which can reduce the loss of granularity and feature resolution for more accurate 3D information recovery.**Module2: RGB-D fusion module** is specifically designed to fully leverage the features of RGB and predicted depth images for nutrition estimation. This module integrates the proposed multi-scale fusion network and cross-modal attention block (CAB). The multi-scale fusion network effectively enriches the intricate semantic features of fine-grained food images, while the CAB further enhances the complementarity of RGB and depth features.

To achieve accurate nutrition estimation, both food recognition and portion estimation are essential. The depth prediction module plays a crucial role in generating depth maps, which serve as the basis for precise portion estimation. The multi-scale fusion network combines multi-level feature maps to generate a high-resolution, strongly semantic feature map, which enhance the extensively recognition of the co-occurring food items in the images. DPF-Nutrition is designed to estimate the nutritional value of food in an end-to-end manner; thus, CAB integrates the complementary information from RGB images and predicted depth maps, enabling the cross-modal information to collaborate in nutrition estimation.

In general, our DPF-Nutrition consists of depth prediction module and RGB-D fusion module, as shown in Figure 3. Specifically, we utilized the ResNet101 network [23] as the backbone of our DPF-Net. In depth prediction module, we employed the Dense Prediction Transformer (DPT) [24] to generate a predicted depth map. The generated depth map was then combined with the RGB image as the input of RGB-D fusion module. In RGB-D fusion module, we extracted RGB features and depth features using separate ResNet networks. The extracted features at the same ResNet layer were then fed into the Cross-modal Attention Block (CAB). The CAB suppressed redundant features within each modality and combined the complementary features. Then, the fused RGB-D features of different resolutions were fed into the third ResNet network and fused from shallow layers to deep layers. This multi-scale fusion process ensured that the final feature map contained comprehensive and detailed information. The feature map was finally processed through global average pooling and a separate multi-task head to estimate the nutritional composition.

#### 2.2.1. The Depth Prediction Module

Depth prediction plays a crucial role in computer vision since it enhances the understanding and perception of real 3D scenes. The depth prediction model typically follows a pattern comprising an encoder and a decoder [25,26,27]. The encoder extracts the features from the input images, while the decoder combines the features from the encoder and converts them into the ultimate depth prediction. The choice of the backbone network for the encoder is essential since the feature information that is lost in encoder cannot be recovered in following decoder. Compared with the depth information of other outdoor scenes or large objects, the depth information of food is more intricate due to its varied geometry and abundant texture. The loss of resolution and granularity during feature extraction in the encoder is prone to causing distortion in the final depth prediction. However, the convolutional neural network [23,28] inevitably loses granularity in deeper stages, since the increase in receptive field and abstraction of features are reliant on the downsampling operation. In comparison, Vision Transformer (ViT) [29] abandons the downsampling operation and maintain the global receptive field of all stages. This makes it more suitable to be used as an encoder for fine-grained food images.

The depth prediction module comprised a transformer encoder and a convolutional decoder. The transformer encoder was responsible for extracting the bag-of-words representation, while the decoder reconstructed the bag-of-words representation into image-like features of different scales. Finally, image-like features were combined into the depth estimation. The structure of DPT is showed in Figure 4a.

Specifically, the input images were abstracted as two-dimensional feature vectors by ResNet-50 network to satisfy the input format of transformer. The feature vectors were then combined with a trainable position embedding to retain positional information. The resulting sequence of embedding vectors, referred to as tokens, were fed into transformer encoder to extract image feature information. Transformer block comprised multi-head self-attention layer, multi-layer perceptron (MLP), two layers of LayerNorm (LN) and residual connections. The structure of transformer encoder is shown in Figure 4b. Multi-head attention enabled transformer encoders to learn representations of both global and local image features. This capability empowered Vision Transformers (ViTs) to capture features at multiple scales, eliminating the requirement for traditional convolutional networks’ gradual downsampling operations. This advancement significantly contributed to maintaining feature resolution and granularity, which are crucial for accurate depth prediction. Given that the resolution of the input image is H × W, after transformer encoders, we obtained a set of tokens *t* = {$t_{0}$⋯$t_{N_{p}}$}, $t_{n}$∈$R^{D}$, where $N_{p}$ = $\frac{H\times W}{p^{2}}$, *D* refers to the dimension of tokens and *p* is the sampling rate of ResNet50.

Then, the $N_{p}$ tokens were spatially concatenated into an image-like feature map through placing each token based on the information of the position embedding:(1)$$Concatnate:R^{N_{p}\times D}\to R^{\frac{H}{p}\times \frac{W}{p}\times D}$$

We resampled the image-like representations into specific size $\frac{H}{s}\times \frac{W}{s}$ with $D^{{}^{\prime }}$ dimensions:(2)$$Resample:R^{\frac{H}{p}\times \frac{W}{p}\times D}\to R^{\frac{H}{s}\times \frac{W}{s}\times D^{{}^{\prime }}}$$

Finally, we employed a RefineNet-based decoder [30] to progressively combine the feature maps at different resolutions and generate the depth prediction. More detailed implementation of DPT can be obtained in the paper [24].

#### 2.2.2. RGB-D Fusion Module

RGB-D fusion is widely applied in various visual tasks including image classification [31], food intake detection [32] and food nutrition assessment [15]. The essence of RGB-D fusion is to combine the complementary information from RGB and depth images to produce more informative and enhanced features. RGB images and depth images pertain to distinct modalities, featuring fundamentally distinct information. Thus, straightforward concatenating or summing cannot fully exploit the cross-modal features. To address this issue, some methods explored improving the efficiency of RGB-D fusion through feature enhancement.

We summarized existing enhancement methods for RGB-D fusion in nutrition estimation into two paradigms: enhancement–fusion [14] and fusion–enhancement [15], as is shown in Figure 5. The specific enhancement techniques can be various. For example, Vinod et al. [14] employed normalization techniques to address disparities in the feature space while Shao et al. [15] employed the balanced feature pyramid and the convolutional block attention module to enhance the fused features. Although these two paradigms could enhance the efficiency of cross-modal fusion, they have certain limitations. The fusion–enhancement method focused on independently enhancing each single-modal feature, without considering the modal complementarity. As a result, this approach might lead to the loss of useful information. In contrast, the enhancement–fusion method enhanced the features after fusion, which might result in residual redundant information from each modal. Differently from the existing methods, we proposed a “fusion–enhancement–fusion” paradigm. Firstly, we performed a straightforward fusion of RGB and depth features to acquire cross-modal interaction information which is then used to enhance both of the single-modal features. Finally, the enhanced single-modal features were combined to generate the fused RGB-D features. This method can effectively remove redundant information from single-modal features while retaining complementary information from cross-modal features. Following this fusion paradigm, we propose a novel cross-modal attention block (CAB) for RGB-D fusion, which is demonstrated in Figure 6. By utilizing the CAB, we enhanced informative features and filter out redundant ones to obtain the most informative fused features. This enables our model to accurately prioritize regions with high nutritional value, rather than solely focusing on specific food items or larger portions.

Specifically, after the depth prediction module, the RGB images together with estimated depth images were input into the feature extraction networks to generate hierarchical features, denoted as RGB features {*R*${}_{i}$}${}_{i=0}^{4}$ and depth features {*D*${}_{i}$}${}_{i=0}^{4}$. The RGB and depth feature maps, with the same resolution, were fed into CAB to generate the complementary fused representation. The structure of CAB is shown in Figure 6. The CAB explicitly built the feature correspondences among different modalities based on the channel and spatial attention vectors of additive features, which emphasized crucial features and suppressed redundant ones. The re-calibrated features were concatenated to obtain complementary cross-modality features. Specifically, given two input features *R*${}_{i}$ and *D*${}_{i}$, we first added the two features pixel by pixel along the channel dimension. Then, we processed the additive features in two branches to obtain channel attention vectors and spatial attention vectors. In channel attention branch, we employed global average pool to obtain global channel descriptor. Then, the global channel descriptor was fed into a 1 × 1 convolution with BN and ReLU to increase non-linearity. Finally, it went through a sigmoid activation function to produce the channel attention vector; the procedure can be defined as:(3)$$CA=Sig(Conv_{1\times 1}\left(GAP\left(R_{i}⊕D_{i}\right)\right))$$
where *GAP*(·) denotes the global average pooling, ⨁ denotes the element-wise addition, *Sig*(·) represents the sigmoid function, and $Conv_{1\times 1}$ indicates a convolutional layer with 1 × 1 kernel size, followed by BN and ReLU.

In spatial attention branch, we calculated the average value for all pixels of the additive feature map along the channel dimension to obtain spatial descriptor. Then, we applied a 3 × 3 convolution with BN and ReLU to smooth the spatial descriptor. Finally, the spatial attention vector was obtained by passing it through a sigmoid activation function; the procedure can be defined as:(4)$$SA=Sig(Conv_{3\times 3}\left(Mean\left(R_{i}⊕D_{i}\right)\right))$$
where *Mean*(·) denotes mean function along the channel dimension. The attention weights of channel and spatial dimensions enhanced the correlation and complementarity of the RGB and depth features. Based on cross-modal attention, the enhanced RGB feature map and depth feature map were obtained. The enhanced features were then further concatenated and fed into a 1 × 1 convolution to obtain the complementary RGB-D features *C*${}_{i}$; the procedure can be defined as:(5)$$C_{i}=Conv_{1\times 1}\left(Concat\left(R_{i}⊗CA⊗SA,D_{i}⊗CA⊗SA\right)\right)$$
where *Concat*(·) denotes cross-channel concatenation and ⨂ indicates pixel-wise multiplication.

After the proposed CAB, we obtained the cross-modal features {*C*${}_{i}$}${}_{i=0}^{4}$. To further enhance semantic features, we adopted a multi-scale fusion network to combine the local detailed information of low-level features and the global context information of the high-level features, progressively. This enabled our model to extensively capture the co-occurring food items, especially the small objects in the images, resulting in accurate nutrition estimation. We used the first cross-modal feature map *F*${}_{0}$ as the input of the feature fusion network; the feature was then refined by a ResNet convolutional block and combined with the next cross-modal feature map *C*${}_{1}$ to generate the fused feature map *F*${}_{1}$. We repeated this operation at the next stage. In this way, we combined the cross-modal features at different scales. The procedure can be defined as:(6)$$F_{i}=C_{i}⊕Res_{i}\left(C_{i-1}\right)$$
where *Res*${}_{i}$ indicates the i-th ResNet convolutional block.

Through the multi-scale fusion network and CAB, we fully utilized the complementary RGB and depth information to generate the final feature representation *F*${}_{4}$. Then, it was fed into a full connected (FC) layer and five multi-task FC heads (with dimensions 2048 and 1, respectively) to generate the estimated nutrition values.

#### 2.2.3. Loss Function

As a multi-task learning model, our DPF-Nutrition predicts the contents of calories, mass and three essential macronutrients of food (fat, carbohydrates and protein). For each subtask, L1 loss was used to measure the bias between the estimated nutritional values and ground-truth ones, which can be defined as:(7)$$L_{cal}=\frac{1}{N}\sum_{i=1}^{N}\left|y_{cal}-y_{cal}^{{}^{\prime }}\right|$$
where *L*${}_{cal}$ denotes the subtask loss of calories, y${}_{cal}$ indicates the estimated value of calorie and y${}_{cal}^{{}^{\prime }}$ represents the ground-truth calorie value. The loss of other subtasks follow this equation.

The scale of the subtask losses are various, which can cause some tasks to dominate other tasks during the training phase. The geometric loss combination [33] is invariant to the scale of individual losses, thereby maintaining a balanced approach towards subtask losses of varying scales. Thus, we used the geometric loss strategy as our loss function, which can be defined as:(8)$$L_{total}=\sqrt[{5}]{L_{cal}L_{mass}L_{fat}L_{carb}L_{protein}}$$
where *L*${}_{tatal}$ denotes the overall loss function.

### 2.3. Evaluation Metrics

In this paper, we adopted the two evaluation metrics of mean absolute error (*MAE*) and percentage of mean absolute error (*PMAE*), which are defined as:(9)$$\;MAE=\frac{1}{N}\sum_{i=1}^{N}\left|y_{i}-y_{i}^{{}^{\prime }}\right|$$
(10)$$\;PMAE=\frac{\mathrm{MAE}}{\frac{1}{N}\sum_{i=1}^{N}y_{i}}$$
where *y* is the estimated nutrient value while *y*${}^{{}^{\prime }}$ is the ground-truth nutrient value. Caloric values are measured in standard kilocalorie units, while the other three nutrient values are measured in grams. A higher level of accuracy in nutrient estimation is achieved when the MAE and PMAE values for evaluation are lower.

In addition, to evaluate the accuracy of depth estimation, we adopt the three evaluation metrics of absolute relative error (*AbsRel*), root mean squared error (*RMSE*) and accuracy with threshold, which are defined as:(11)$$\;AbsRel=\frac{1}{N}\sum_{i=1}^{N}\frac{\left|d_{i}-d_{i}^{{}^{\prime }}\right|}{d_{i}}$$
(12)$$\;RMSE=\sqrt{\frac{1}{N}\sum_{i=1}^{N}{\left|d_{i}-d_{i}^{{}^{\prime }}\right|}^{2}}$$
(13)$$\;Accuracy=\%\;of\;d_{i}\;s.t.\;max\left(\frac{d_{i}}{d_{i}^{{}^{\prime }}}\;,\frac{d_{i}^{{}^{\prime }}}{d_{i}}\right)=\delta <thr$$
where *d* represents the ground-truth depth value while *d*${}_{i}$ represents the predicted depth value. A higher level of accuracy in depth prediction is achieved when the AbsRel and RMSE values are lower and accuracy is nearer to 1.

## 3. Results

### 3.1. Experimental Detail

All the experiments were conducted on a 24G NVIDIA GTX 3090 GPU. To maintain the experimental fairness, the same setup was utilized for all experiments. For the training of the depth prediction module, we resized the input images to have a long side of 384 pixels and train on random square crops of size 384. The process not only meets the input size requirement of the vision transformer but also serves as an augmentation technique to enhance the model’s generalization capabilities. The encoder network was initialized with ImageNet1K pre-trained weight while the decoder network was initialized randomly. We utilized the Adam optimizer [34] with an initial learning rate of 1 × 10${}^{-5}$ and implemented a cosine annealing strategy for learning rate decay. The learning rate declined to 1 × 10${}^{-6}$ after cosine annealing. We trained the model for 60 epochs with a batch size of 8. For the training of the RGB-D fusion module, the input images were resized to 336 × 448 pixels to reduce memory consumption while maintaining the original aspect ratio. Image augmentation methods including center cropped and random horizontal flip were applied to training images to enhance the model’s generalization capabilities. The backbone network was initialized with Food2K [35] pre-trained weight. Food2K is a large dataset for fine-grained food recognition that offers features specifically suitable for transfer learning in food-related vision tasks. We chose the Adam optimizer with an initial learning rate of 5 × 10${}^{-5}$ and implemented an exponential decay strategy for updating the learning rate, with a decay rate set to 0.98. We trained the model for 150 epochs with a batch size of 8. The optimization techniques, including optimizer selection, learning rate strategies and batch size are all carefully chosen based on experimental experiences, in order to achieve the best possible results.

### 3.2. Backbone Comparison

It is essential to select a suitable backbone networks for the model. In this section, we made a comparison among several widely used convolutional neural networks (CNNs) and the recently popular vision transformer [29]. According to Table 2, ResNet101 achieved better performance, with the best mean PMAE of 20.9%. Therefore, we used ResNet101 [23] as the backbone network for our DPF-Nutrition, unless otherwise specified.

### 3.3. Depth Prediction Analysis

To evaluate the performance of the Dense Prediction Transformer (DPT) in food images, we compare it with three depth prediction methods based on fully convolutional networks. To ensure a fair comparison, we consistently maintained the experimental settings and dataset splits across all methods. We compare the methods including: DPT [24], FCRN [37], UNet [26] and HRNet [38]. The experimental results are shown in Table 3. Due to the vision transformer’s capability of preserving image resolution and granularity throughout the encoding process, the Dense Prediction Transformer (DPT) achieved the best performance in depth estimation.

Furthermore, in order to assess the influence of depth map quality on nutrition estimation, we evaluated the performance of nutrition estimation models that integrated RGB images and depth maps. These depth maps included the ground-truth ones and those generated by DPT, FRCN, UNet and HRNet. All the experiments employed the same RGB-D fusion module and experimental settings. The result is shown in Table 4. The results of the actual depth maps were best, with a mean PMAE of 17.2%. And a better performance of nutrition estimation was achieved when the quality of the predicted depth maps was better.

### 3.4. Cross-Modal Fusion Analysis

To evaluate the effectiveness of our proposed CAB, we compared it with two RGB-D fusion methods. The result is shown in Table 5. These two methods followed the cross-modal fusion paradigms demonstrated in Figure 5, referred to as “fusion–enhancement” and “enhancement–fusion”. To ensure fairness in the comparison, we maintained consistency in the specific feature enhancement method, which included spatial attention and channel attention mechanisms. Additionally, the experimental settings and dataset splits remained the same. CAB utilized cross-modal interaction information to enhance single-modal features so as to generate complementary fused features. Among the three methods compared, CAB demonstrated the best performance by following a “fusion–enhancement–fusion” paradigm, resulting in a mean PMAE of 17.8%.

### 3.5. Method Comparison

To evaluate the performance of DFP-Nutrition, we compared it with three representative nutrition estimation methods. To ensure a fair comparison, we consistently maintained the experimental settings and dataset splits across all methods. The experimental results are shown in Table 6. We referred to the methods proposed by Thames et al. [10] and Shao et al. [15] as Google-Nutrition and RGB-D Nutrition, respectively. We compared the methods including: Google-Nutrition, RGB-D Nutrition, Swin-Nutrition [11] and our proposed DPF-Nutrition.

Google-Nutrition was introduced in conjunction with the publication of the Nutrition5k dataset, which was widely regarded as the baseline method for nutrition estimation. Google-Nutrition offered two variations: Google-Nutrition-monocular, which utilized monocular images, and Google-Nutrition-depth, which incorporated food images and depth data. Swin-Nutrition achieved the state-of-the-art performance in nutrition estimation based on monocular images. Swin-Nutrition employed Swin-Transformer [39] as the backbone and used a feature fusion module to obtain discriminative feature representation to improve the accuracy of nutrition estimation. It should be noted that the results of Swin-Nutrition in this paper differed from the original ones published in [11]. This is because we replicated their method to avoid the data inconsistency caused by their experimental data cleaning. RGB-D Nutrition achieved the state-of-the-art results in nutrition estimation based on RGB-D images. RGB-D Nutrition employed balanced feature pyramid [16] and convolutional block attention module [17] to perform effective RGB-D fusion to improve the accuracy of nutrition estimation.

As demonstrated by the experimental results, DPF-Nutrition showed advantages in monocular image-based nutrition estimation. In comparison to Google-Nutrition-monocular and Swin-Nutrition, the mean PMAE improved by 11.3% and 2.6%, respectively. This improvement highlights the effectiveness of the deep prediction module in enhancing the accuracy of nutrition prediction by incorporating 3D information. Furthermore, when compared to the state-of-the-art RGB-D method, DPF-Nutrition demonstrated a competitive performance with a 0.7% improvement in the mean PMAE. Considering the potential impact caused by the bias between predicted depth maps and actual depth maps, the improvement underscored the effectiveness of our exploration of RGB-D fusion for nutrition estimation. Overall, our DPF-Nutrition exhibited a highly competitive performance when compared to existing methods.

### 3.6. Ablation Study

To validate the effectiveness of the proposed modules, comprehensive ablation studies were conducted on the various components comprising our DPF-Nutrition. The experimental results are shown in Table 7. The baseline was the RGB stream that used only a ResNet101 network to estimate nutrients from RGB images. Similarly, depth stream used a single network to estimate nutrients from depth images generated by the depth prediction module. Model (c) and Model (d) employed simple feature vector concatenation and the proposed multi-scale fusion network, respectively, to integrate the RGB stream and depth stream. Model (e) represented the complete DPF-Nutrition which incorporates the CAB on the basis of Model (d).

According to Table 7, Model (a) achieved a mean PMAE of 20.9%, whereas model (b) achieves a mean PMAE of 38.6%. The results demonstrated that relying solely on depth information alone cannot accurately estimate nutrients. Model (c) concatenated the RGB and depth features at the last layer of the feature extraction network, and achieves a mean PMAE of 19.6%, which is a 1.5% improvement compared to Model (a). The result demonstrated the effectiveness of complementing RGB images with estimated images for nutrition estimation and validated the efficacy of our depth prediction module. The improvements achieved by Model (d) and Model (e) evaluated the effectiveness of the proposed multi-scale fusion network and Cross-modal Attention Block (CAB), respectively. Model (e) achieved an improvement over Model (c), with a 1.8% decrease in mean PMAE, as well as reductions of 4.8 kCal in calories MAE and 3.8 g in mass MAE. The results of the ablation experiment demonstrated that all of the depth prediction module, multi-scale fusion network and CAB can effectively improve the accuracy of nutrient prediction.

### 3.7. Visualization Analysis

To visually showcase the efficacy of our method, we began by visualizing the performance of the depth prediction module. As depicted in Figure 7, our depth prediction module successfully recovered the depth information and smoothed out the noise in the original depth.

Next, we employed Grad-Cam [40] to visualize the RGB-D fusion module, which generated a heat map highlighting the regions of interest (ROIs) identified by the model. This visualization allowed for a more intuitive understanding of how our method leveraged food images to make accurate predictions. The visualization results of certain dishes and the corresponding nutrition facts are shown in Figure 8. The regions that the model focuses on are marked in red, while the opposite regions are marked in blue. As illustrated in the visualization, our model demonstrated a focused attention on specific regions for each nutrient task. For instance, when estimating the fat and protein content in dish_156278816, the model exhibited a strong focus on the pork and fish components. Conversely, when estimating the carbohydrate content, the model primarily emphasized the rice and corn elements. This indicated that our model effectively captured relevant visual cues and assigned appropriate importance to different regions based on the specific nutrient being estimated.

## 4. Discussion

Daily nutrient intake is essential for people’s health. According to previous studies, it has been found that a moderate caloric restriction (ranging from 18% to 30%) can lead to various positive effects on the health of obese patients, including a reduction in visceral fat, an improvement in insulin sensitivity and a decrease in the risk of developing metabolic diseases [41,42]. Furthermore, the results of a long-term experiment revealed that, after therapy, the weight of participants of obesity class II decreased by 22.4% while their calorie and carbohydrate intake decreased by 18.3% and 15.6%, respectively [43]. Although our DPF-Nutrition demonstrated an improvement in nutrition estimation, it is important to note that there were still errors in estimating the contents of calories, mass, protein, fat and carbohydrates, which reached 14.7%, 10.6%, 20.2%, 22.6% and 20.7% respectively. These errors may be disadvantageous, especially in the context of dietary monitoring for medical purposes.

In order to investigate the limitations of our model, we examined the test samples that yielded poor results. We discovered that the most significant errors primarily originated from three categories of images. The first category includes images with food stacking and covering. For instance, in the case of dish_1560367980, where the low-calorie spinach significantly obscured the high-calorie pizza, the estimated calorie value was 31.6% lower than the actual value. The second category comprises images containing minuscule components that are imperceptible, such as oil and sugar. For example, in the case of dish_1562617939, where a large amount of olive oil was added to the dish, the calorie PMAE was up to 52.5%, while the fat PMAE was dramatically 68.3%. The third category comprises images with food items that are rarely encountered in the training data. For example, in the case of dish_1562617703, which involved the uncommon ice-cream in the Nutrition5k dataset, the calorie PMAE reached up to 39.5%, while the carbohydrate PMAE reached up to 57.4%. It is evident that food stacking, imperceptible components and insufficient data are the primary factors that restrict the performance of DPF-Nutrition. When we removed these samples in the experiments, the performance of DPF-Nutrition in the estimation of calories, mass, protein, fat and carbohydrates reached 13.4%, 10.2%, 21.2%, 19.2% and 18.9%, respectively. The mean PMAE improved by 1.2%. Although removing these challenging samples can result in improved results, the dietary scenarios containing food stacks and imperceptible ingredients cannot be ignored, as they are real-life situations. In addition, the impact of scarcity of datasets on food nutrition estimation is significant. Deep learning techniques rely heavily on the dataset; however, the diversity and scale of the existing nutrition datasets are far from sufficient for practical application. Furthermore, due to the high local and cultural characteristics of food, the transferability between datasets of different cuisines are relatively weak. For example, the model trained on the Western food dataset cannot be directly used for nutrition estimation of Chinese food. Generally, Chinese dishes offer a wide range of variety, with complex ingredients and richer cooking methods, which can pose more challenges in vision-based nutrition estimation.

In the future, we will focus on overcoming the problems of food stacking and imperceptible components for more accurate nutrition estimation. One potential solution for food stacking involves detecting the stacked portion in the food image, conducting structural analysis on this segment and ultimately calculating nutrition adjustments. Furthermore, we are committed to creating an extensive public food dataset to facilitate comprehensive dietary assessment. Given the limited transferability across food datasets, our goal is to construct a specialized dataset, specifically focusing on a Chinese breakfast dataset. This dataset will be compact yet of high quality, ensuring its practical usability in daily applications. In addition to further improving our algorithms, we are also ambitious to integrate DPF-Nutrition into practical applications. Many existing dietary tracking apps such as MyFitnessPal, MyDietCoach and FatSecret require some manual input, such as recalling what you ate or weighing the food. In contrast, our proposed DPF-Nutrition achieves a fully automated process from capturing an image to estimating its nutritional content. The practical application of DPF-Nutrition holds immense potential in alleviating user burden, reducing costs and enhancing accuracy in dietary monitoring. However, the practical application of DPF-Nutrition should consider additional issues such as computational complexity and image quality. Our current design primarily focuses on improving accuracy, without taking into account the computational complexity of the model. Yet, in practical scenarios, the computational complexity directly affects resource utilization and running speed, which are closely associated with user experience. Therefore, in the next stage, we need to address the problem of reducing the model’s complexity while maintaining accuracy. Moreover, factors like angles, distances and lighting conditions during the image capture process can all impact image quality. It is crucial to find solutions to mitigate the impact of low image quality on nutrition estimation, as this is another practical issue that needs consideration in the application of DPF-Nutrition.

We strongly believe that the study of automated nutritional assessment holds great promise. By automating dietary assessment, individuals can receive real-time feedback on the nutritional content of their meals, thereby increasing their awareness and understanding of their dietary choices. This empowers individuals to make informed decisions about their food intake and ultimately adopt healthier eating habits. The convenience, affordability and efficiency of automated dietary assessment make it suitable for meeting the dietary monitoring needs of diverse populations. It is our hope that our study will contribute to the advancement of automated dietary assessment, ultimately enhancing dietary education and improving public health outcomes.

## 5. Conclusions

In this paper, we proposed our DFP-Nutrition for dietary assessment, aiming to develop an automated, cost-effective and precise method for nutrition estimation. Our proposed method offered a novel approach by predicting the depth map from a monocular image and incorporating the recovered 3D information with RGB food images to improve nutrition estimation. To assess the effectiveness of our method, we performed experiments on the Nutrition5k dataset and compared its performance with that of state-of-the-art image-based nutrition estimation methods. The results clearly demonstrated the effectiveness of our proposed method, showcasing its competitive performance when compared to other existing methods. In the future, we envision the widespread utilization of automated vision-based nutrition estimation methods in our daily lives, making a significant contribution to improving dietary education and public health.

## Figures and Tables

**Figure 1 foods-12-04293-f001:**
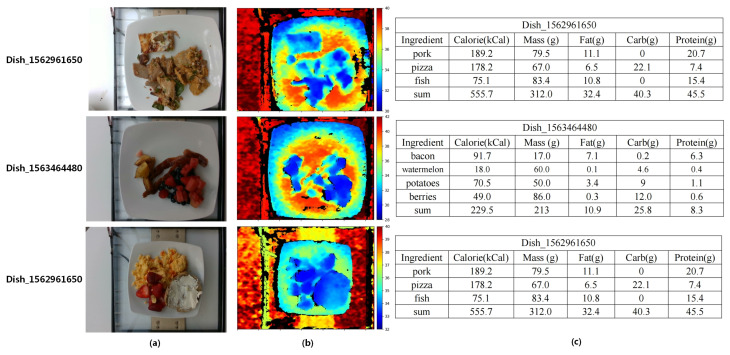
The example images from Nutrition5k dataset. (**a**) RGB images. (**b**) Depth maps. (**c**) Nutritional annotations.

**Figure 2 foods-12-04293-f002:**
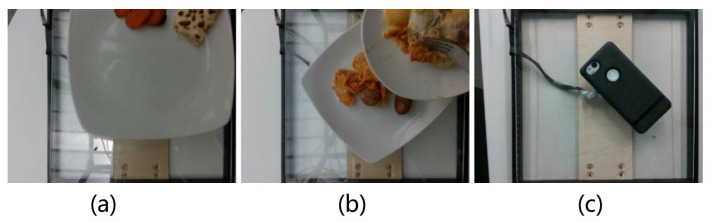
Incorrect image samples. (**a**) Food is not fully incorporated in the image. (**b**) Dishes are overlapping. (**c**) Non-food image.

**Figure 3 foods-12-04293-f003:**
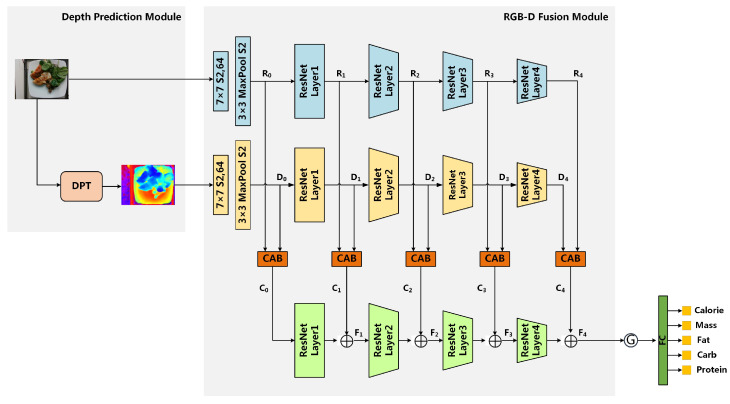
The overall framework of our DPF-Nutrition, which consists of depth prediction module and RGB-D fusion module. We adopt depth prediction transformer (DPT) to generate the predicted depth map. We design a cross-modal attention block (CAB) to extract and integrate the complementary features of RGB and depth images. ⨁ indicates element-wise addition, Ⓖ denotes global average pool.

**Figure 4 foods-12-04293-f004:**
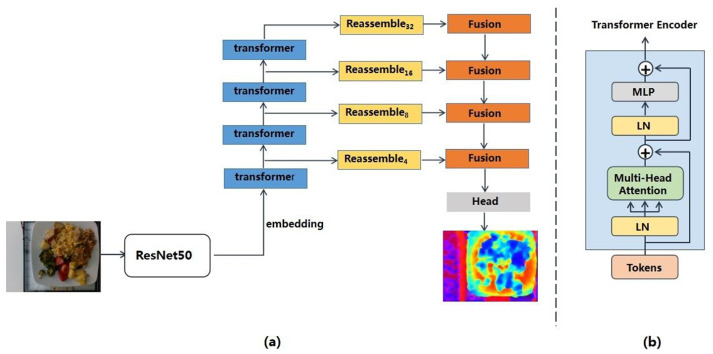
(**a**) The structure of depth prediction module. The input image is transformed into feature vectors by ResNet-50 feature extractor and consequently embedded into two-dimensional tokens. The tokens are then fed into transformer encoder. The tokens from different transformer stages are reassembled into image-like feature maps at various resolutions. Finally, the image-like feature maps are fused progressively to generate the depth prediction. (**b**) The structure of transformer encoder. ⨁ indicates element-wise addition.

**Figure 5 foods-12-04293-f005:**
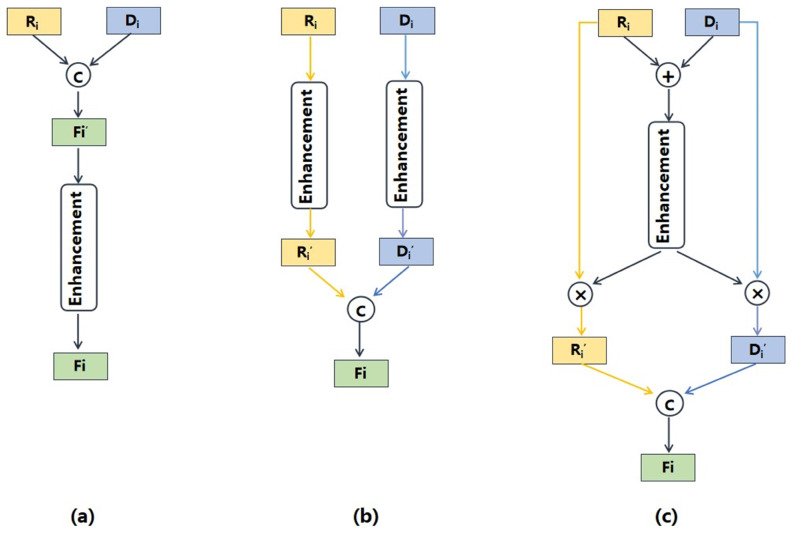
The structures of the RGB-D fusion paradigms. (**a**) Fusion–enhancement. (**b**) Enhancement–fusion. (**c**) Our proposed. ⨁ denotes element-wise addition, ⨂ indicates pixel-wise multiplication, Ⓒ represents cross-channel concatenation.

**Figure 6 foods-12-04293-f006:**
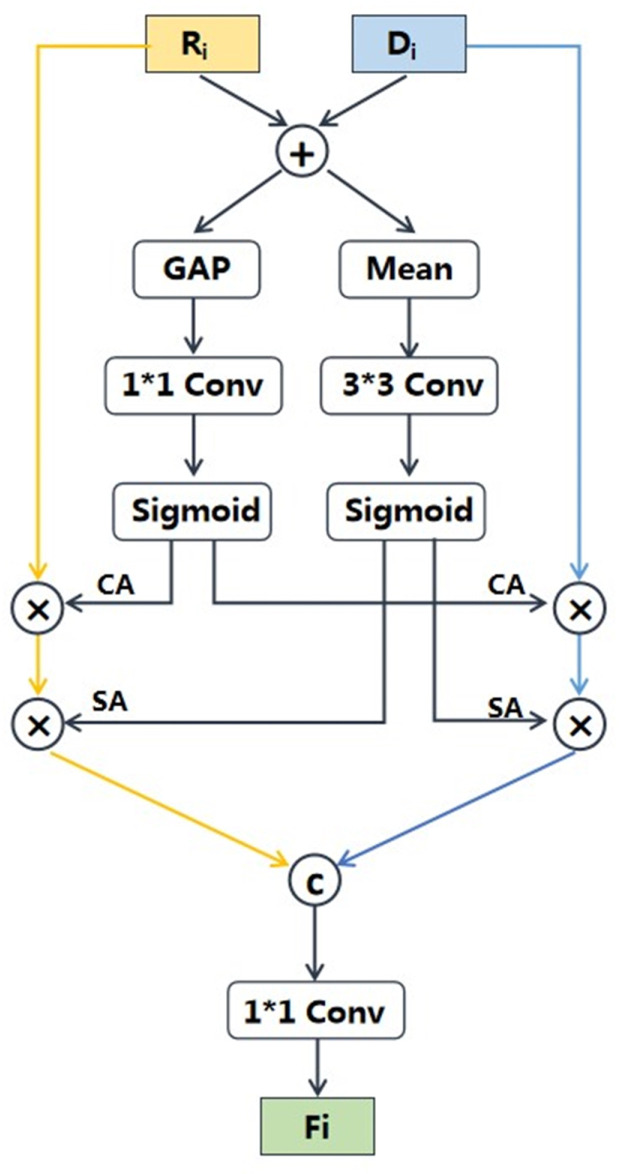
The structure of CAB. GAP indicates global average pooling, ⨁ denotes element-wise addition, ⨂ indicates pixel-wise multiplication, Ⓒ represents cross-channel concatenation, Mean represents mean function along the channel dimension.

**Figure 7 foods-12-04293-f007:**
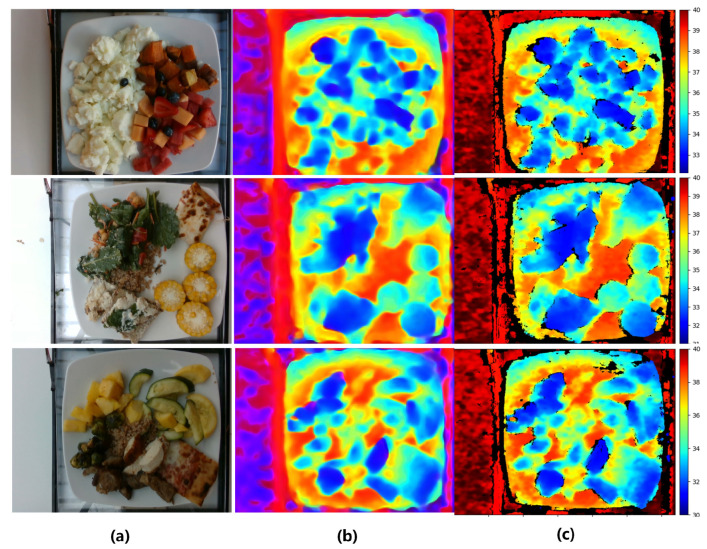
The sample results of the depth estimation. (**a**) RGB images. (**b**) Estimated depth maps. (**c**) Actual depth maps.

**Figure 8 foods-12-04293-f008:**
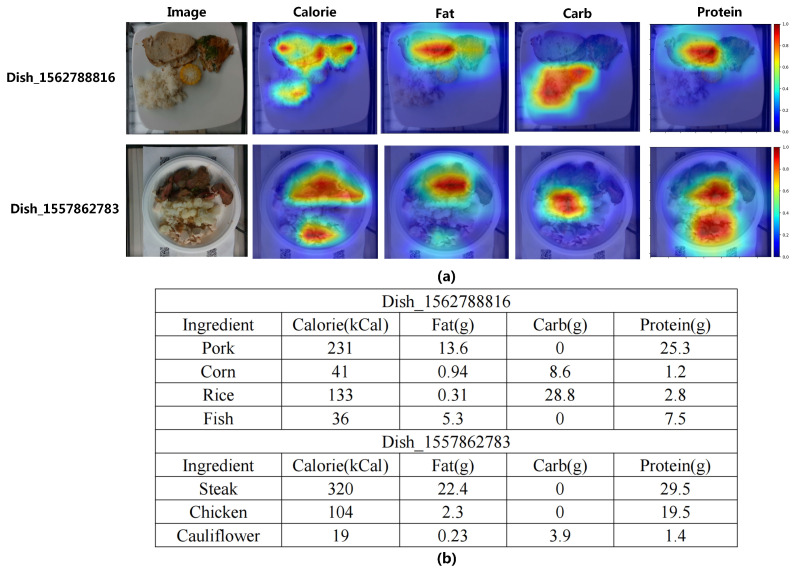
The visualization results. (**a**) The ROI heat-maps of different nutrients. (**b**) The nutrition facts.

**Table 1 foods-12-04293-t001:** Summary of nutrition datasets. Diversity represents the number of unique dishes and size represents the number of the images.

Dataset	Diversity	Size	Annotation	Depth
MenuMatch [19]	41	646	Calories	N
ECUSTFD [20]	160	2978	Volume and Mass	N
Fang et al. [21]	3	45	Calories	N
Ando et al. [22]	3	270	Calories	Y
Nutrition5k [10]	5066	3490	Calories, Mass and Macronutrients	Y

**Table 2 foods-12-04293-t002:** Comparison of the performance of different backbones. The best results were highlighted in bold.

Methods	Calories MAE (kCal)/PMAE (%)	Mass MAE (g)/PMAE (%)	Fat MAE (g)/PMAE (%)	Carb MAE (g)/PMAE (%)	Protein MAE (g)/PMAE (%)	Mean PMAE (%)
ViT [29]	52.4/20.4	32.5/16.3	3.78/29.4	5.72/28.9	5.72/28.9	24.6
VGG16 [28]	47.9/18.6	29.2/14.6	3.61/28.0	5.18/26.2	4.71/26.8	22.8
InceptionV3 [36]	46.2/18.0	28.4/14.2	**3.24/25.1**	5.1/25.8	4.37/24.8	21.6
ResNet50 [23]	46.9/18.2	27.7/13.8	3.61/28.0	**4.51/22.8**	4.56/25.9	21.7
ResNet101 [23]	**46.0/17.9**	**27.2/13.6**	3.42/26.5	4.56/23.0	**4.31/24.5**	**21.1**

**Table 3 foods-12-04293-t003:** Comparison of the performance of different depth prediction methods.The best results were highlighted in bold.

Method	δ>1.25	$\delta >1.25^{2}$	$\delta >1.25^{3}$	AbsRel	RMSE
UNet [26]	0.573	0.771	0.860	0.661	0.211
FCRN [37]	0.634	0.799	0.877	0.529	0.189
HRNet [38]	0.692	0.864	0.906	0.387	0.162
DPT [24]	**0.743**	**0.893**	**0.942**	**0.322**	**0.128**

**Table 4 foods-12-04293-t004:** Comparison of the performance of nutrition estimation using depth maps from different sources.The best results were highlighted in bold.

Depth Source	Calories MAE (kCal)/PMAE (%)	Mass MAE (g)/PMAE (%)	Fat MAE (g)/PMAE (%)	Carb MAE (g)/PMAE (%)	Protein MAE (g)/PMAE (%)	Mean PMAE (%)
UNet [26]	41.6/16.2	22.6/11.3	3.29/25.5	4.43/22.4	4.01/22.8	19.6
FRCN [37]	40.7/15.8	22.7/11.3	3.17/24.6	4.28/21.6	3.87/22.0	19.1
HRNet [38]	39.3/15.3	21.6/10.8	3.01/23.3	4.13/20.9	3.72/21.2	18.3
DPT [24]	37.9/14.7	21.2/10.6	2.92/22.6	4.09/20.7	3.56/20.2	17.8
Depth sensor	**36.5/14.2**	**20.4/10.2**	**2.76/21.4**	**4.08/20.6**	**3.48/19.8**	**17.2**

**Table 5 foods-12-04293-t005:** Comparison of the performance of different cross-modal fusion methods.The best results were highlighted in bold.

Method	Calories MAE (kCal)/PMAE (%)	Mass MAE (g)/PMAE (%)	Fat MAE (g)/PMAE (%)	Carb MAE (g)/PMAE (%)	Protein MAE (g)/PMAE (%)	Mean PMAE (%)
Enhancement–Fusion	39.1/15.2	22.6/11.3	3.03/23.5	4.14/20.9	3.80/21.6	18.5
Fusion–Enhancement	39.3/15.3	21.6/10.8	3.01/23.3	4.13/20.9	3.72/21.2	18.3
CAB	**37.9/14.7**	**21.2/10.6**	**2.92/22.6**	**4.09/20.7**	**3.56/20.2**	**17.8**

**Table 6 foods-12-04293-t006:** Comparison of the performance of different methods.The best results were highlighted in bold.

Input	Methods	Calories MAE (kCal)/ PMAE (%)	Mass MAE (g)/ PMAE (%)	Fat MAE (g)/ PMAE (%)	Carb MAE (g)/ PMAE (%)	Protein MAE (g)/ PMAE (%)	Mean PMAE (%)
RGB-D images	Google-Nutrition-depth [10]	47.6/18.8	40.7/18.9	**2.27/18.1**	4.6/23.8	3.7/20.9	20.1
	RGB-D Nutrition [15]	38.5/15.0	21.6/10.8	3.0/23.5	4.43/22.4	3.69/21.0	18.5
Monocular images	Google-Nutrition-monocular [10]	70.6/26.1	40.4/18.8	5.0/34.2	6.1/31.9	5.5/29.5	29.1
	Swin-Nutrition [11]	41.5/16.2	27.5/13.7	3.21/24.9	4.32/21.8	4.47/25.4	20.4
	DPF-Nutrition (ours)	**37.9/14.7**	**21.2/10.6**	2.92/22.6	**4.09/20.7**	**3.56/20.2**	**17.8**

**Table 7 foods-12-04293-t007:** Comparison with different ablation settings.The best results were highlighted in bold.

Index	Model	Calories MAE (kCal)/ PMAE (%)	Mass MAE (g)/ PMAE (%)	Fat MAE (g)/ PMAE (%)	Carb MAE (g)/ PMAE (%)	Protein MAE (g)/ PMAE (%)	Mean PMAE (%)
(a)	RGB Stream	46.0/17.9	27.2/13.6	3.42/26.5	4.56/23.0	4.31/24.5	21.1
(b)	Depth Stream	83.5/32.5	44.7/22.3	6.29/48.8	8.08/40.8	8.53/ 8.5	38.6
(c)	(a) + (b) + direct fusion	42.7/16.7	25.0/12.5	3.17/24.6	4.31/21.8	3.97/22.6	19.6
(d)	(a) + (b) + multi-scale fusion	40.7/15.8	24.8/12.4	3.01/23.3	4.13/20.9	3.88/22.1	18.9
(e)	(d) + CAB	**37.9/14.7**	**21.2/10.6**	**2.92/22.6**	**4.09/20.7**	**3.56/20.2**	**17.8**

## Data Availability

The data used to support the findings of this study can be made available by the corresponding author upon request.

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
