# Peer review of "DPF-Nutrition: Food Nutrition Estimation via Depth Prediction and Fusion"

_foods, 2023, doi:10.3390/foods12234293_

Round 1

Reviewer 1 Report

Comments and Suggestions for Authors

Dear Authors,

In general, the manuscript is almost ready; the structure of the work is clear and complete. After reading the paper, it is clear that the authors are very experience in operations on database objects and modules of operation. All tested methods should be useful in collective catering and industrial production of dishes and especially dedicated meals intended for people following a special diet. However, I recommend as proposal for future research to use elaborated methods to the estimation of real dishes or to estimate separate ingredients of dishes.    

In my opinion, it is necessary to verify the study using real dishes of known value of each ingredient (weight, calorie, fats, carb, and protein) to have estimation of accuracy of the methods you used.

See my comments below, which most of them will be helpful in a future study:

1.      Lines 113, 159-160, 346-347, 361-362. – It concern all figures: 1, 4, 6, and 7, where are submitted color pictures representing: image from Nutrion5k, color image of depth prediction, ground-truth depth and estimated depth, and hit maps of different nutrients. You should add color bars, that it must indicate range of changes and scale, on which may be represented values of indicated properties or ingredient.

2.      Lines 113 – Why are submitted the example of image the same as from main page of  Nutrient5k website: https://github.com/google-research-datasets/Nutrition5k

Did you have any others images to show in the paper?

3.      Generally, I don’t understood why in many papers, like in this paper authors use RGB system to estimate pictures. It is well know, that RGB system is use to describe source of light, and CMY is use to describe pictures, from which the light is reflected.

4.      Any statistical methods were not applied. For example, in table 4. There are two single dishes separately only. 

I also suggested some papers that can complete and improve literature review concerning on other methods of food quality estimation: “How to identify roast defects in coffee beans based on the volatile compound profile”, “Some physical and nutritional quality parameters of storage apple”, “Hyperspectral imaging coupled with multivariate analysis and artificial intelligence to the classification of maize kernels”, “Fresh Broccoli in Fortified Snack Pellets: Extrusion-Cooking Aspects and Physical Characteristics”, “Effect of supplementation of flour with fruit fiber on the volatile compound profile in bread”.

Comments on the Quality of English Language

Nomenclature concerning with food products, and the manuscript is written clear and in understandable language for reader.

Author Response

Thank you very much for taking the time to review this manuscript. We have carefully revised the original version according to the suggestions of the reviewers. Please find the response letter  in the attachment and the corresponding revisions highlighted changes in the re-submitted files.

Reviewer 2 Report

Comments and Suggestions for Authors

Dear Yuzhe Han, Qimin Cheng, Wenjin Wu, and Ziyang Huang, 

Subject: Major Revision Required for Manuscript Titled "DPF-Nutrition: Food Nutrition Estimation via Depth Prediction and Fusion"

I have had the opportunity to thoroughly review your manuscript entitled "DPF-Nutrition: Food Nutrition Estimation via Depth Prediction and Fusion", which outlines an innovative approach to food nutrition estimation by integrating depth prediction with RGB-D fusion techniques. I commend your team for tackling the intricacies of dietary health monitoring and for striving to enhance the accuracy of food portion estimation using advanced deep learning methodologies.

Your paper's aim to reconcile the convenience of monocular image-based methods with the precision of RGB-D approaches is both ambitious and highly relevant in the context of the rapidly evolving field of automated dietary assessment. The introduction of a novel RGB-D fusion module and the employment of a multi-scale fusion network and Cross-modal Attention Block (CAB) are indeed impressive aspects of your work.

However, after careful consideration, it has been determined that the paper would benefit substantially from further enhancement. Therefore, I am recommending a major revision. The following points highlight areas that, if addressed, could improve the robustness and impact of your manuscript:

  1. Technical Rigor: The depth prediction model's methodology necessitates a more detailed explanation, especially regarding its ability to replicate the precision of actual depth sensors. Additionally, a comparative analysis with state-of-the-art depth prediction algorithms could solidify your method's standing.
  2. Experimental Validation: While the results on the Nutrition5K dataset are promising, validation across additional datasets could strengthen the generalizability of your findings. It would also be beneficial to provide a more comprehensive discussion of any limitations encountered during your experiments.
  3. Comparative Analysis: A deeper comparative study with current monocular and RGB-D methods, including both quantitative and qualitative assessments, would provide a clearer understanding of your method’s advantages and potential trade-offs.
  4. Practical Implications: A discussion on how DPF-Nutrition could be integrated into existing dietary tracking applications and the associated user experience implications would be highly valuable. This includes considerations around user cost, practicality, and ease of use in everyday scenarios.
  5. Reproducibility: To aid in reproducibility and facilitate further research, consider providing access to code repositories, detailed algorithmic parameters, and any additional resources that could assist researchers in replicating your study.
  6. Broader Impact: Elaborate on the potential societal impact of your method, specifically its role in public health, dietary education, and its adaptability to diverse populations with varying dietary needs.

We believe that addressing these suggestions will not only solidify the scientific contribution of your work but also enhance its appeal to a broader audience. We appreciate the effort and expertise invested in your research thus far and anticipate that the revisions will lead to a substantial improvement in the manuscript.

Please submit your revised manuscript along with a point-by-point response to the comments outlined above. We look forward to receiving your enhanced manuscript and are optimistic about its potential contribution to the field of food nutrition estimation.

Thank you for considering these points. Should you have any questions or require further clarification, please do not hesitate to contact me.

Sincerely,

Please see my Review Report in the attached file!

Comments on the Quality of English Language

The English language necessitates minor editorial polishing.

Dear Yuzhe Han, Qimin Cheng, Wenjin Wu, and Ziyang Huang, 

Subject: Major Revision Required for Manuscript Titled "DPF-Nutrition: Food Nutrition Estimation via Depth Prediction and Fusion"

I have had the opportunity to thoroughly review your manuscript entitled "DPF-Nutrition: Food Nutrition Estimation via Depth Prediction and Fusion", which outlines an innovative approach to food nutrition estimation by integrating depth prediction with RGB-D fusion techniques. I commend your team for tackling the intricacies of dietary health monitoring and for striving to enhance the accuracy of food portion estimation using advanced deep learning methodologies.

Your paper's aim to reconcile the convenience of monocular image-based methods with the precision of RGB-D approaches is both ambitious and highly relevant in the context of the rapidly evolving field of automated dietary assessment. The introduction of a novel RGB-D fusion module and the employment of a multi-scale fusion network and Cross-modal Attention Block (CAB) are indeed impressive aspects of your work.

However, after careful consideration, it has been determined that the paper would benefit substantially from further enhancement. Therefore, I am recommending a major revision before reconsideration for publication. The following points highlight areas that, if addressed, could improve the robustness and impact of your manuscript:

  1. Technical Rigor: The depth prediction model's methodology necessitates a more detailed explanation, especially regarding its ability to replicate the precision of actual depth sensors. Additionally, a comparative analysis with state-of-the-art depth prediction algorithms could solidify your method's standing.
  2. Experimental Validation: While the results on the Nutrition5K dataset are promising, validation across additional datasets could strengthen the generalizability of your findings. It would also be beneficial to provide a more comprehensive discussion of any limitations encountered during your experiments.
  3. Comparative Analysis: A deeper comparative study with current monocular and RGB-D methods, including both quantitative and qualitative assessments, would provide a clearer understanding of your method’s advantages and potential trade-offs.
  4. Practical Implications: A discussion on how DPF-Nutrition could be integrated into existing dietary tracking applications and the associated user experience implications would be highly valuable. This includes considerations around user cost, practicality, and ease of use in everyday scenarios.
  5. Reproducibility: To aid in reproducibility and facilitate further research, consider providing access to code repositories, detailed algorithmic parameters, and any additional resources that could assist researchers in replicating your study.
  6. Broader Impact: Elaborate on the potential societal impact of your method, specifically its role in public health, dietary education, and its adaptability to diverse populations with varying dietary needs.

We believe that addressing these suggestions will not only solidify the scientific contribution of your work but also enhance its appeal to a broader audience. We appreciate the effort and expertise invested in your research thus far and anticipate that the revisions will lead to a substantial improvement in the manuscript.

Please submit your revised manuscript along with a point-by-point response to the comments outlined above. We look forward to receiving your enhanced manuscript and are optimistic about its potential contribution to the field of food nutrition estimation.

Thank you for considering these points. Should you have any questions or require further clarification, please do not hesitate to contact me.

Sincerely,

Please see my Review Report in the attached file!

Author Response

Thank you very much for taking the time to review this manuscript. We have carefully revised the original version according to the suggestions of the reviewers. Please find the response letter  in the attachment and the corresponding revisions highlighted in the re-submitted files.

Round 2

Reviewer 2 Report

Comments and Suggestions for Authors

Dear Authors,

The comments have been revised.

Congratulations! 

Comments on the Quality of English Language

English Language fine.